# Municipal Residence Level of Long-Term PM_10_ Exposure Associated with Obesity among Young Adults in Seoul, Korea

**DOI:** 10.3390/ijerph17196981

**Published:** 2020-09-24

**Authors:** Jayeun Kim, Kyuhyun Yoon

**Affiliations:** 1Institute of Health and Environment, Seoul National University, Seoul 08826, Korea; kimjayeun@gmail.com; 2Graduate School of Education, Kyung Hee University, Seoul 02447, Korea

**Keywords:** Community Health Survey, CHS, PM_10_ long-term effect, BMI

## Abstract

Background: long-term effects of ambient pollutants used to be defined in cohort studies using biomarkers. Health effects on young adults from long-term exposure to particulate matters (PM) in residential ambiance have received less attention. Methods: using the data of population-representative aged 19–29 in Seoul, the relationship between obesity and PM_10_ levels of the living district was examined. We defined obesity as Body Mass Index (BMI) 25 kg/m^2^ and more. Survey logistic regression was conducted according to individual residence periods in the current municipality. Individual characteristics were adjusted overall and were age-specific; aged 19–24 and 25–29. Results: study population was 3655 (1680 (46%) men and 1933 aged 19–24 (52.9%)) individuals. Relationship between length of residence in municipalities with a greater level of PM_10_ from 2001–2005 and obesity was increased over the residing period; 10 years ≤ (odds ratio (OR) 1.071, 95% confidence interval (CI) 0.969–1.185), 15 years ≤ (1.120, 1.006–1.247), and 20 years ≤ (1.158, 1.034–1.297) in aged 19–29. Age-specific effects showed slight differences. Conclusions: Although PM_10_ levels are currently decreasing, higher levels of PM_10_ exposure in the residential area during the earlier lifetime may contribute to obesity increase among young adults.

## 1. Introduction

Worldwide obesity has nearly tripled since 1975 and overweight or obese populations accounted for 39% and 13%, respectively, out of those aged 18 years and over in 2016 [1]. In the International Classification of Diseases 11th Revision (ICD-11), obesity is classified as code 5B81. Code 05 is endocrine, nutritional, or metabolic diseases [2]. Obesity is a chronic, relapsing, multi-factorial, and neurobehavioral disease [3]. Therefore, it is necessary to consider a wide spectrum of risk factors including environmental conditions such as ambient air pollution as well as sociodemographic and behavioral factors to understand the obesity epidemics.

As one of the environmental factors related to obesity, particulate matters (PM) were investigated for various chronic conditions and acute episodes [4,5,6], age groups [7,8,9,10], gender [4], regions [11], and socioeconomic statuses such as education level [12,13], household income [13,14,15,16,17,18], or occupational characteristics [12]. They are mostly based on a cohort study to figure out the long-term effect of PM. In those examined, some studies suggested increasing associations but the impact of PM on obesity remains mixed [19,20].

Obesity prevalence is increasing among younger generations worldwide and a similar trend has been shown in Korea over the past two decades. Obesity among young adults is an emerging health issue because the early onset of obesity with various chronic conditions will be a potential risk for rising expenditure. However, the relationships between obesity and PM among young adults were not clearly defined because they are less likely to be prioritized since they are not considered to be less vulnerable than children or the elderly.

Long-term effects of ambient pollutants were usually analyzed in cohort studies. Those results’ ascertainment focused on individual exposures measured with their biomarkers. Various studies have shown the long-term effects of PMs in early life on childhood obesity [21,22]. However, in terms of public health, to make an initiative for community intervention or action in the residing community, we would find a more efficient and economical approach to enhance study feasibility. For this reason, we adopted an ecological approach to examine the association between community-based long-term exposure of PM_10_ and obesity among young adults using data from the municipality level of PM_10_. We used data from the municipality level of PM_10_ and individual health survey outcomes. In this study, we tested a hypothesis on whether long-term exposure of higher PM_10_ in the municipality where each study subject lives is related to a higher probability of obesity among young adults. In order to examine the relation between the level and exposure period of PM_10_ and obesity, we tested different exposure periods and time based on the municipality residing period for individuals and set a separate time window such as 5 or 10 years in aggregating PM_10_ levels.

## 2. Materials and Methods

### 2.1. Study Subjects and Variables

We used the 2017 Community Health Survey (CHS) data. The survey was conducted since 2008, nationwide in 250 municipal health centers, by the Center for Disease Control and Prevention (CDC) of Korea. The participants are randomly sampled adults (19 years ≤) under the stratified systematic sampling by municipalities. Combined individual and household sampling weights were provided with the data.

The CHS collected health-related factors from several domains such as demographic factors, personal height and weight, health behaviors and socioeconomic information under the cross-sectional study design annually.

This study’s populations were young adults aged from 19 to 29 years old dwelling in 25 municipalities in the Seoul metropolis. Since exposure to pollutants in childhood has been known to affect obesity, we wanted to limit the age to the age group that includes the past 20 years as a child (≤12). The survey question for the residence period (RP) in the current municipality had five categorical responses; less than 5 years (RP < 5), 5–10 years (5 ≤ RP < 10), 10–15 years (10 ≤ RP < 15), 15–20 years (15 ≤ RP < 20), and greater than 20 years (20 ≤ RP). In order to explore the relation with obesity associated with higher PM_10_ exposure for a long time, the study subjects were restricted to the subjects under 30 years old. Consequently, considering air pollutant data available from 2001, we enrolled study subjects aged between 19 and 29. Their demographics, socioeconomic status (SES), and health behaviors including the information of residence period in the current municipality were selected for analysis.

Air pollutant data were collected among 25 municipalities from their own monitoring stations from the year 2001 to 2017 by the Korea Environment Corporation (Air Korea, Seo-gu, Incheon, Korea).

PM_10_ (µg/m^3^) level was analyzed as the form of annual average concentrations. The presence of particulate matter (PM) is valid to sort according to the fraction size. PM_10_ are particles of aerodynamic diameter <10 µm (PM_10_), and accordingly PM_2.5_ are particles of aerodynamic diameter <2.5 µm (PM_2.5_). Accordingly, PM_2.5_ is a much finer size than PM_10_ and is a subset of PM_10_ in size. Using the hourly measured PM_10_ levels in each municipality site, we calculated the daily mean and averaged out daily values into the annual level according to each municipality between 2001 and 2017 (Figure 1). Furthermore, we aggregated annual PM_10_ levels into different terms of periods such as 5 years, 10 years, and 15 years. Municipalities were grouped into two levels: ‘high’ and ‘low’ levels of long-term PM_10_, using the median value as a cutoff onto each periodic average of PM_10_.

### 2.2. Statistical Analysis

The CHS data are collected from population-representative samples. Therefore, secondary obesity, due to endocrine disorders, hypothalamic disorders, or some congenital conditions and medications for those clinical conditions, were not considered in our study. BMI was calculated using personal weight and height and usually classified into four categories: underweight (<18.5), normal weight (18.5≤ and <25), overweight (25≤ and <30), and obese (30≤) [23]. It is well known that body fat percentages in Asians are 3–5% higher than other races [24]. In this study, we used the weight and height collected individual basis to calculate BMI and followed the Korean classification under WHO guidelines for the Asia-Pacific region [25,26]. Accordingly, obesity was defined as BMI 25 or greater.

Initially, we investigated the distribution of demographics, SESs, and health behaviors according to the residence period in the current municipality. Secondly, we assessed the association of population using logistic regression analysis for survey data with a complex sampling design between obesity and individual characteristics including demographics, SES, and health behaviors. Finally, under survey logistic regression analysis, we estimated the long-term effect of municipal PM_10_ by fitting the aggregated PM_10_ levels after adjusting sociodemographic variables and health behaviors., Furthermore, the age-specific effect was explored among those aged 19–24 and 25–29 in estimating associations between obesity and municipal PM_10_ level under the fitting model adjusted demographic factors listed ahead.

All results related to the association are presented as odds ratios (OR) with 95% confidence intervals (CIs). All procedures were conducted using SAS version 9.4 (SAS Institute, Inc., Cary, NC, USA) and all figures were modeled using R 4.0.2 (The Comprehensive R Archive Network: http://cran.r-project.org). All statistical tests were 2-sided and *p* < 0.05 was considered statistically significant.

### 2.3. Ethics

This study was approved as exempt by the Institutional Review Board of Seoul National University (IRB No. E1611/001–003). Informed consent was obtained when survey interviewers were collecting CHS data from the individual participants. The data were subsequently anonymized and de-identified before analysis.

## 3. Results

### 3.1. Descriptive Statistics among Study Subjects

The total participants were 3655 (men 1680 (46.0%) and aged 19–24 1933 (52.9%)) (Table 1). Approximately 85% of the participants had an educational background with university or higher levels and nearly 60% took part in the labor market. The average sleep hours per day during a recent week were distributed as < 6 (540 (14.8%)), 6–7 (1140 (312%)), and 7 ≤ (1975 (54.0%)). Current smokers were 665 (18.2%). The majority of the study populations did not perform moderate or higher physical activity (2723 (74.5%)) and their obesity prevalence was 23.1% (845 subjects). The national and Seoul regional trends have increased between 2008 and 2017 while annual aggregated PM_10_ levels of Seoul have decreased between 2001 and 2017 (Figure 2).

### 3.2. Associations between Obesity and Individual Factors

The association between obesity and demographic characteristics were explored and the results of multiple regression analysis were presented in Table 2. Women were less associated with obesity compared (OR 0.493, 95% CI 0.450–0.541) to men. The impact of education level did not show a consistent trend. Inactive labor market participation was less associated with obesity compared (OR 0.998, 95% CI 0.914–1.089) to the active. Sleeping hours of seven or greater per day showed less obesity (OR 0.880, 95% CI 0.787–0.983) than shorter sleepers. Former smoking gave increased association (OR 1.245, 95% CI 0.997–1.555) compared to never smoking for obesity outcome. Residence period in a current municipality was shown varied but not statistically significant associations according to the length of residing.

### 3.3. Associations between Obesity and Municipal Long-Term PM_10_ Level

We fitted, by turns, municipal PM_10_ levels with aggregated period according to the length of residing and age group with the adjustment of individual sociodemographic factors and health behaviors (Table 3).

Among the municipalities with high PM_10_ levels during 2001–2005, the various but increased associations with obesity were observed among overall study subjects aged 19–29 according to the residence period: 10 years ≤ (OR 1.071, 95% CI 0.969–1.185), 15 years ≤ (OR 1.120, 95% CI 1.006–1.247), and 20 years ≤ (OR 1.158, 95% CI 1.034–1.297). Additionally, decreased associations between obesity and levels of PM_10_ from 2006–2010 were mainly observed regardless of the residence period.

Effect Modification by age was detected in assessing the effect of PM_10_ levels from 2001–2005 by the residence period: 20 years ≤ (OR 1.229, 95% CI 1.081–1.397) among those aged 19–24, and 10 years ≤ (OR 1.208, 95% CI 1.054–1.383) and 15 years ≤ (OR 1.213, 95% CI 1.062–1.385) among the other age group 25–29. Meanwhile, the association with obesity consistently decreased for the high PM_10_ municipalities. Between 2006 and 2010, among those aged 25–29 with statistical significance regardless of residence period: 10 years ≤ (OR 0.825, 95% CI 0.723–0.943) and 15 years ≤ (OR 0.826, 95% CI 0.728–0.937), and 20 years ≤ (OR 0.825, 95% CI 0.725–0.939).

In order to figure out the modification effect according to the residence period, associations for the specified residence periods were calculated and the results were presented separately (Appendix A). Relations were jagged according to the varied long-term community-based PM_10_ levels within the same residence period and age-specific groups.

## 4. Discussion

### 4.1. Principal Findings

The association between obesity and long-term PM_10_ levels among young adults was rarely studied and defined ambiguously. Since the level of PM_10_ has been decreasing during the past two decades, the short-term effects of PM_10_ have not been distinct in obesity and other factors such as diet and physical exercise could work with population obesity more closely. On the other hand, the long-term approach of PM_10_ exposure from early life is possibly different from the exposed PM_10_ level. This study supported that long-term exposure of high PM_10_ in the community could increase the risk of obesity. The study results showed that participants living longer in municipalities with higher PM_10_ levels were more likely to be overweight or obese than those living with the lower level of PM_10_ in the residence area. Therefore, in the application of a community-based approach for interventions or building strategies to reduce obesity prevalence, both health behaviors and environmental factors should be considered and discussed at the same time.

### 4.2. Associations between Obesity and Individual Factors

At an individual level, the World Health Organization (WHO) suggested engagement in regular physical activity to reduce overweight and obesity [1]. In our study, the study participants who had not engaged in moderate or vigorous physical activity showed that they were less likely to be in an obesity group. As long as the current study data were collected based on a cross-sectional survey design, it could be interpreted that obesity subjects more frequently participated with regular exercise to reduce weight.

Shorter sleep and poor sleep quality might contribute to the development of obesity in epidemiological studies using self-reported cross-sectional studies including young adult populations [27,28,29]. Similar to the previous results, our study indicated that greater than 7 h of sleep per day had a decreased association with obesity compared to the participants with less than 6 h of sleep per day.

In smoking behavior, former smokers are more likely to be obese while less association was detected in current smokers compared to never smokers. The positive relations between obesity and smoking were reported with higher smoking rates in obesity participants among young adults [30] and middle-aged adults [31]. However, considering the general perception that smoking may reduce weight gain, especially among young women, smoking cessation intervention may need to consider weight management support.

In the middle-aged, effect modifications of obesity, such as adverse associations between particulate matters and cardiovascular health effects were suggested in several cohort studies [27,32,33]. Our study subjects were restricted to those in their twenties and so these participants would be healthier and fewer would have chronic conditions. Therefore, modification under comorbidity conditions was not considered in this study.

### 4.3. Associations between Obesity and Environmental Factors

Over several decades, the worldwide trend of obesity has increased among young adults [1]. Although decreasing trends of PM_10_ are observed in some areas, PM_10_ exposure is still preserved as one of the candidates of obesity risk factors. In our study, we examined PM_10_ as a representative of the urban area’s outdoor pollutants and detected that the relation increased when exposed to higher PM_10_ levels in earlier lifetime among young adults. Ambient PM_10_ or PM_2.5_ levels are highly related to traffic. Urbanized cities like Seoul are more likely to have greater traffic-related air pollution. To figure out the consistency of this finding, several studies were referred to and these studies were conducted to investigate the relationship between traffic-related air pollution and obesity, especially for children [20,34] or young adults [10]. In addition, among primary school-aged children, exposure to ambient air pollution was associated with overweight and obesity at home and school [35]. From those studies, an increased association between PM level and obesity was detected.

In our study, we set a hypothesis that the population exposed to a higher level of PM_10_ is more likely to be obese compared to those dwelling areas with a lower level of PM_10._ It was examined using municipal PM_10_ levels. In Table 3, despite some statistically significant odds ratios, the effects of PM_10_ levels, remained low between 1.120 and 1.229. We can interpret these small figures as follows: in terms of health determinants, there are several dimensional factors to determine the individuals’ health outcomes [36]. It is known that health outcome is determined according to demographic factors such as age, sex, or genetic conditions. It is further suggested that environmental conditions such as ambient pollutants are furthest from the core of health determinants.

In Table 1, there are several variables: smoking, drinking, and education that seem to have a greater influence on obesity prevalence in a statistical approach. Odds ratio (OR) figures were small because the effects of relatively peripheral atmospheric variables were observed after adjusting for variables (individual characteristics) that more directly affect individual obesity in terms of health determinants. However, although the OR figures are relatively small, the authors regard our study results as worthwhile to support decision making to improve environmental conditions including air pollutant level which is prospectively changeable.

On the other hand, although the association between obesity and community PM_10_ level was increased for some exposure period from 2001–2005, the association was inversed during the exposure period 2006–2010 among high PM_10_ level’s municipalities. The decreased association was more pronounced among those aged 25–29 than 19–24, regardless of residence period.

In the 2000s, the characteristics of moving households in Seoul were mainly the thirties in household heads’ age and professionals in job classification, with 37.0% and 40.1%, respectively [37,38]. In other words, a group with relatively high socioeconomic status in their thirties was moved to Seoul at that time. The thirties are the age of marriage and birth of the first child in Korean culture. Young professionals flow into Seoul, where the cost of living is higher than in other areas and more opportunities for promotion and a rich educational infrastructure for their children. Therefore, it can be inferred that their young children had good health behavior and health conditions due to a high socioeconomic level rather than the environmental impact of the immigrated region.

Calculating participants’ age retrospectively for each period, for instance, 29-year-old participants were 18 and 22 years old in 2006 and 2010, respectively. Using the same backward calculation, the median aged group, those aged 24 years old and the least aged group, 19 years old in the year 2017 were aged 13–17 and 8–12 in 2006 and 2010, respectively. Applying that calculation, this study subjects’ age ranged from 2 to 12 years old. Those age differences may pull the effect variability and may accompany the importance of early life exposures because of the expectation of more hazardous health outcomes. Due to the importance of early life exposure, the Human Early Life Exposome (HELIX) studied a prospective cohort to measure multiple environmental exposures during early life (pregnancy and childhood) to figure out child health outcomes [39]. Furthermore, a similar approach was preceded to describe the key principles from the literature regarding the relevance of these principles to early life neurotoxic exposures [40].

The toxic mechanism inducing obesity, caused by PM, begins with pro-inflammatory cytokine release followed by systemic inflammation. Then it collects inflammatory cells into adipose and other tissues [41]. The toxic reactions of high levels of air pollution (including PM_2.5_ and ultrafine particles) are known to lead to reactions such as lipogenesis, lipolysis, and hypothalamic inflammation due to excessive intake or high-fat diet [42].

Apart from air pollutants, the availability of neighborhood greenness [43,44] or physical activity facilities [45,46,47] for physical activity were considered as beneficial environmental factors related to obesity. We reviewed the annual statistics of greenness areas and local autonomous sports facilities. Through this approach, it was concluded that annual change was negligible in both variables and not to include them as another environmental factor. Instead, among the demographic variables, information on the frequency of physical activity during the week was used as a confounding variable. For those who exercise regularly for weight management, it is common to use private institutions around their homes, workplaces, or schools. However, this study did not consider the frequency of subjects’ use of private institutions because they could not be determined.

### 4.4. Study Limitations

Our study has some limitations on air pollutant data and health outcomes. This study aimed to access the long-term effect of PM_10_ on obesity among young adults dwelling in a community. First of all, the health data only provide the current status of obesity and no additional information for the incidence time point. Accordingly, the long-term association is limited to assure the extension to the causal association. Secondly, in accessing the relationship between obesity and PM_10_, our study was limited in the accuracy of PM_10_ exposure amount and period for individual study participants due to study design in data collection protocol. However, it strengthens the previous study results that earlier exposure in a lifetime to higher levels of PM_10_ has driven those to become obese with a higher probability. Finally, the association with obesity may vary by the size of the PM fraction. Although several studies compared the fraction size of PM and defined the association including coarse particles (PM_10-2.5_), fine particulate matters (PM_2.5_), and even ultrafine (PM_0.1_), it was not accessed in the current research because only recent years of PM_2.5_ measurements were available and no data were available for a finer fraction. The particle constituents are known to be varied according to the fraction size of PM and the relation with health may vary [48,49,50]. Recently, ultrafine particles (PM_0.1_) were explored and suggested for the inflammatory responses [51]. Although finer fractions of PM were not explored in this study, those finer particles such as PM_2.5_ or PM_0.1_ are subsets of PM_10._ Since examining the health effect related to finer particles is not available in a current ecological approach, it is agreed that additional research is required after collecting finer PM data.

## 5. Conclusions

As part of the participation and efforts of various organizational or individual forms, levels of particulate matters are decreasing in some parts of the world. However, there are still signals that the level of particulate matters exceeds WHO standards. Although currently, PM_10_ levels are decreasing in Seoul, Korea, this study suggested that higher levels of ecological PM_10_ exposure during earlier lifetime may contribute to increasing individual obesity among young adults. In order to understand the increasing obesity epidemic, a wide spectrum of risk factors including environmental exposures and exposure time and period should be considered. Applying timely intervention for the overall population in order to make tailoring policies to improve physical health from obesity is necessary to approach community-based environmental improvement.

## Figures and Tables

**Figure 1 ijerph-17-06981-f001:**
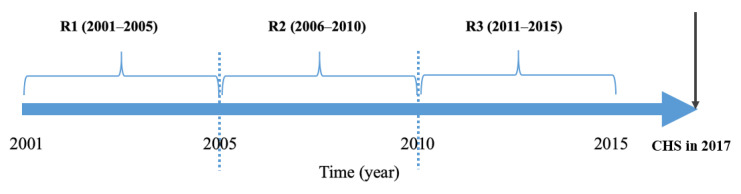
Study design applying a periodic window for exposure of particulate matters and residence period using Community Health Survey data in 2017, Korea.

**Figure 2 ijerph-17-06981-f002:**
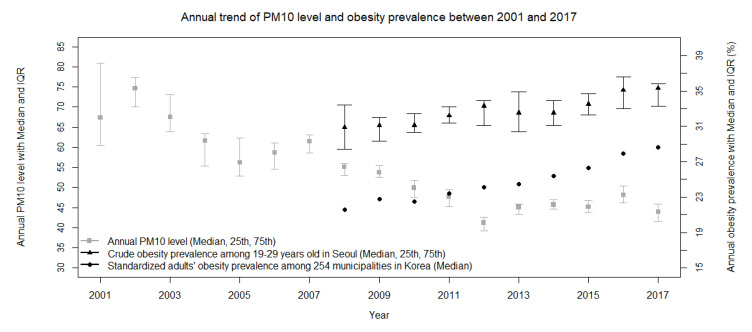
Annual trend of particulate matter (PM_10_(μg/m^3^)) levels and obesity prevalence, annual level of PM_10_ was calibrated using hourly measured level of PM_10_ among 25 municipalities in Seoul between 2001 and 2017 and presented as median value with 25th and 75th. Crude obesity prevalence among study populations aged between 19 and 29 years old in Seoul was presented as a form of median with 25th and 75th. Standardized obesity prevalence among adults aged equal or greater than 19 years old was presented as a form of median and this information was derived from Statistics of Community Health Survey, 2008–2017, published by the Center for Disease Control and Prevention.

**Table 1 ijerph-17-06981-t001:** Descriptive characteristics and obesity prevalence among the study population.

Variable	Category	Proportion/ Prevalence, *n* (%)		
		Overall	Obese ^1^	Chi-Squared *(p-*Value*)*
Total		3655 (100.0)	845 (23.1)	
Sex	Men	1680 (46.0)	620 (36.9)	184.64 (<0.0001)
	Women	1975 (54.0)	225 (11.4)	
Age group	19–24	1933 (52.9)	416 (21.5)	0.20 (0.6547)
	25–29	1722 (47.1)	429 (24.9)	
Education	Master’s course	192 (5.3)	43 (22.4)	761.44 (<0.0001)
	University	2929 (80.1)	655 (22.4)	
	High school or less	532 (14.6)	147 (27.6)	
	Missing	2 (0.1)	0 (0.0)	
Labor market participation	Active	2199 (60.2)	505 (23.0)	32.22 (<0.0001)
	Inactive	1455 (39.8)	340 (23.4)	
	Unknown	1 (0.0)	0 (0.0)	
Average sleep hours (hours per day)	6-	540 (14.8)	141 (26.1)	150.82 (<0.0001)
	6–7	1140 (31.2)	272 (23.9)	
	7+	1975 (54.0)	432 (21.9)	
Moderate or vigorous physical activity rate	Yes	927 (25.4)	257 (27.7)	129.02 (<0.0001)
	No	2723 (74.5)	587 (21.6)	
Smoking	Never	2802 (76.7)	547 (19.5)	421.50 (<0.0001)
	Former	188 (5.1)	68 (36.2)	
	Current	665 (18.2)	230 (34.6)	
Drinking	Never	237 (6.5)	44 (18.6)	1254.58 (<0.0001)
	Former	158 (4.3)	34 (21.5)	
	Current	3260 (89.2)	767 (23.5)	
Residence period (years) in a current community (RP)	RP < 5	1120 (30.6)	237 (21.2)	252.67 (<0.0001)
	5 ≤ RP < 10	377 (10.3)	87 (23.1)	
	10 ≤ RP < 15	402 (11.0)	97 (24.1)	
	15 ≤ RP < 20	434 (11.9)	106 (24.4)	
	20 ≤ RP	1322 (36.2)	318 (24.1)	

^1^ 25 ≤ BMI (Body Mass Index).

**Table 2 ijerph-17-06981-t002:** Results of logistic regression of obesity with variables considered among young adults in Seoul, Korea, 2017.

Variable	Category	Odds Ratio (95% Confidence Interval) ^1^
Sex	Men	1.00
	Women	0.493 (0.450–0.541)
Age	19–24	1.00
	25–29	1.076 (0.987–1.172)
Education level	Master’s course	1.00
	University	0.937 (0.806–1.090)
	High school or less	1.073 (0.899–1.281)
Labor market participation	Active	1.00
	Inactive	0.998 (0.914–1.089)
Average sleep hours (hours per day)	6-	1.00
	6–7	0.948 (0.837–1.073)
	7+	0.880 (0.787–0.983)
Moderate or vigorous physical activity rate	Yes	1.00
	No	0.960 (0.880–1.047)
Smoking	Never	1.00
	Former	1.245 (0.997–1.555)
	Current	0.926 (0.793–1.081)
Drinking	Never	1.00
	Former	1.187 (0.888–1.588)
	Current	1.000 (0.825–1.211)
Residence period in a current community (RP)	RP < 5	1.00
	5 ≤ RP < 10	1.096 (0.850–1.414)
	10 ≤ RP < 15	0.961 (0.769–1.201)
	15 ≤ RP < 20	1.127 (0.902–1.408)
	20 ≤ RP	0.982 (0.848–1.136)

^1^ Odds ratio was calculated with those who have obesity (25 ≤ BMI) against those with under or normal weight in survey logistic regression.

**Table 3 ijerph-17-06981-t003:** Associations between obesity and exposure to high level of PM_10_ according to residence period among young adults in Seoul, Korea, 2017.

Period of PM_10_ Level ^1^	Study Population		PM_10_ Level Median Cutoff	Study Population Residence Period (Year)	Odds Ratio (95% Confidence Interval) ^2^		
	Overall, *n*	Obese, *n* (%)			Age 19–29	Aged 19–24	Aged 25–29
5 years (2001–2005)	2158	521 (24.1)	68.7	10 years≤	1.071 (0.969–1.185)	0.986 (0.881–1.104)	1.208 (1.054–1.383)
	1756	424 (24.2)		15 years≤	1.120 (1.006–1.247)	1.059 (0.932–1.204)	1.213 (1.062–1.385)
	1322	318 (24.1)		20 years≤	1.158 (1.034–1.297)	1.229 (1.081–1.397)	1.112 (0.971–1.273)
5 years (2006–2010)	2158	521 (24.1)	56.6	10 years≤	0.894 (0.808–0.991)	0.945 (0.841–1.061)	0.825 (0.723–0.943)
	1756	424 (24.2)		15 years≤	0.934 (0.838–1.041)	1.040 (0.908–1.191)	0.826 (0.728–0.937)
	1322	318 (24.1)		20 years≤	0.889 (0.792–0.997)	0.954 (0.829–1.097)	0.825 (0.725–0.939)
5 years (2011–2015)	2158	521 (24.1)	46.2	10 years≤	1.037 (0.937–1.147)	0.988 (0.882–1.108)	1.096 (0.960–1.251)
	1756	424 (24.2)		15 years≤	1.031 (0.925–1.149)	0.959 (0.842–1.093)	1.123 (0.986–1.279)
	1322	318 (24.1)		20 years≤	1.019 (0.909–1.142)	0.970 (0.848–1.109)	1.067 (0.933–1.219)

^1^ Annual level of PM_10_ (μg/m^3^) was calibrated using hourly measured level of PM_10_ among 25 municipalities in Seoul between 2001 and 2017. ^2^ Odds ratio was calculated with those who have obesity (25 ≤ BMI) against those with under or normal weight in survey logistic regression. The regression model was fitted with socio-demographic factors and health behaviors described in the Table 1 and Table 2.

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
