# Peer review of "Municipal Residence Level of Long-Term PM10 Exposure Associated with Obesity among Young Adults in Seoul, Korea"

_ijerph, 2020, doi:10.3390/ijerph17196981_

Round 1

Reviewer 1 Report

I have noted minor editorial and language issues and suggest having someone slightly more fluent in English review the language for minor corrections.  Otherwise I had a few issues which I believe need to be corrected.

Line 71 - I saw nothing further in this paper on SES status.  If SES was studied what were the results?

Line 99 - Where is SES reported?  You cannot say you investigated it (especially SES) and then not report the results.  Either remove the statement, which means you must report having no results on this or report it as part of your analysis.  You cannot have it both ways.

Lines 167-170 - I am having difficulty understanding the meaning of this sentence.  It seems to be a minor language issue.  I suggest the authors have someone review their translation into English with knowledge of the intent of what they meant to say.

Lines 250-255 - I expected more of an explanation concerning the reversal of trends between 2001-2005 and 2006-2010. Maybe I have missed it but the putative reasons for that reversal certainly seem to warrant further discussion.

Line 269 - SInce you cite a relationship with both PM10 and PM2.5 you are admitting there is a potential confounder (PM2.5) that is unaccounted for and may itself also correlate with PM10.  How do you propose to disentangle those variables? Is that not a limitation as well? Also, PM0.1 (ultrafine particulate) is reputed to have an inflammatory effect which could also lead to obesity and is not mentioned at all.

Author Response

We sincerely appreciate your sincere comments. With those comments, we re-run some analyzes and pondered how to more meaningfully express our previous results. We tried to edit the manuscript thoroughly and improve its fluency. We hope our revision would meet your expectation.

Thank you again.

Best regards.

Kyuhyun Yoon

Reviewer 2 Report

This study aims to evaluate the long-term exposure of higher PM10 in the living municipality is related to the more probability of obesity among young adults.

The authors concluded that the higher levels of PM10 exposure at the residential area during the earlier life-time may contribute to increasing obesity among young adults.

This article is well written and of clinical interest.

However, several issues should be improved before the consideration for publication.

Major comments

1 Inclusion and exclusion criteria for the participants is unclear in this study.

For instance, was the secondary obesity included in this study?

2 Although PM2.5 may be well known in the world, the readers may be unfamiliar with PM10.

Therefore, some concise explanation concerning PM10 is helpful for the readers.

3 Are the data of other air pollution available? Is the association between PM10 and obesity independent of other substances and environments?

4 In Table 3, were the odds ratios calculated considering the confounding factors, for instance, parameters listed in Table 2?

5 In Table 3, odds ratios are small such as 1.12 to 1.21. What do the authors think of them in the interpretation of the results?

6 What is the plausible underlying mechanism between PM10 and obesity in young adults?

Author Response

(The authors gave the same response as above.)

Round 2

Reviewer 2 Report

The manuscript has been improved according to the comments.

However, some minor issues should be considered.

Major comments
1. Inclusion and exclusion criteria for the participants is unclear in this study. For instance, was the secondary obesity included in this study?
Response: As per the comment, we addressed more information for the inclusion and exclusion criteria for the participants in the manuscript. Regarding the secondary obesity, our study data is from the community-based population-representative survey data. Therefore, the secondary obesity which is based on clinical conditions is not included in our study.

Please describe this issue in the method section because some readers may suspect that the secondary obesity due to pharmacotherpy and other endocline diseases may be included in the study.

5. In Table 3, odds ratios are small such as 1.12 to 1.21. What do the authors think of them in the interpretation of the results?
Response: Thanks for the critical point out in interpreting the association between obesity and PM10 levels. In terms of health determinants, there are several dimensional factors to determine the individuals’ health outcome. It is known that health outcome is determined according to demographic factors such as age, sex, or genetic conditions. It suggests further that environmental conditions such as ambient pollutants are the most far from core of health determinants.
In Table 1, there are several variables; smoking, drinking, and education that seem to have a greater influence on obesity prevalence in a statistical approach. Odds ratio (OR) figures were small because the effects of relatively peripheral atmospheric variables were observed after adjusting for variables (individual characteristics) that more directly affect individual obesity in terms of health determinants. However, although the OR figures are relatively small, authors regard that our study results are worthwhile to support a decision making to improve environmental conditions including air pollutants level which is prospectively changeable.

Again, it may be better to add the explantion about small ORs above. Otherwise, readers may underestimate the results.

Author Response

Dear reviewer,

We appreciated your taking the time again. Our revisions to the previously revised manuscript have been highlighted in green with a new line number.

Best regards.  
